# The Effect of TNF and Non-TNF-Targeted Biologics on Body Composition in Rheumatoid Arthritis

**DOI:** 10.3390/jcm10030487

**Published:** 2021-01-29

**Authors:** Gaelle Vial, Céline Lambert, Bruno Pereira, Marion Couderc, Sandrine Malochet-Guinamand, Sylvain Mathieu, Marie Eva Pickering, Martin Soubrier, Anne Tournadre

**Affiliations:** 1Rheumatology Department, CHU Clermont-Ferrand, 63000 Clermont-Ferrand, France; gaellevial@orange.fr (G.V.); mcouderc@chu-clermontferrand.fr (M.C.); smalochet@chu-clermontferrand.fr (S.M.-G.); smathieu@chu-clermontferrand.fr (S.M.); mepickering@chu-clermontferrand.fr (M.E.P.); 2Biostatistics Unit (DRCI), CHU Clermont-Ferrand, 63000 Clermont-Ferrand, France; clambert@chu-clermontferrand.fr (C.L.); bpereira@chu-clermontferrand.fr (B.P.); 3Rheumatology Department, Université Clermont Auvergne, CHU Clermont-Ferrand, INRAE, UNH UMR 1019, 63000 Clermont-Ferrand, France; msoubrier@chu-clermontferrand.fr

**Keywords:** rheumatoid arthritis, body composition, muscle, fat, biologic, TNF inhibitor

## Abstract

Rheumatoid arthritis (RA) is associated with a decrease in lean mass and stability or even an increase in fat and ectopic adipose tissue. A few data are available on body composition changes under treatment, and data are still controversial. Body composition was assessed before initiation of biologic disease-modifying antirheumatic drug (bDMARD) and after 6 and 12 months of stable treatment. Eighty-three RA patients were included (75% of women, mean age 58.5 ± 10.8 years) of whom 47 patients treated with TNF inhibitor (TNFi), 18 with non-TNF-targeted biologic (Non-TNFi), and 18 with conventional DMARD (cDMARD) alone. In the TNFi group, total lean mass, fat-free mass index, and skeletal muscle mass index significantly increased at 1 year. An increase in subcutaneous adipose tissue (SAT) without change for the visceral or body fat composition was associated. These changes were associated with an improvement in strength and walking test. In non-TNFi or cDMARD groups, no significant changes for body composition or muscle function were observed at 1 year. However, no significant differences for treatment x time interaction were noted between group treatments. In active RA patients starting first bDMARD, treatment with TNFi over 1 year was associated with favorable changes of the body composition and muscle function.

## 1. Introduction

Biologic disease-modifying antirheumatic drugs (bDMARDs) represent a major advance in the management of rheumatoid arthritis (RA). However, RA remains associated with an increased mortality mainly due to of cardiovascular (CV) diseases [1]. In addition to the traditional CV risk factors, inflammation and metabolic disorders such as insulin resistance and alteration in body composition may contribute to CV risk and mortality [2,3,4,5]. Altered body composition, characterized by a decrease in lean mass while fat mass may be preserved or even increased regardless of changes in total body weight, is correlated to disease activity [6] and quality of life [2]. Optimizing body composition may counteract the reduced mobility induced by joint pain and deformity [7]. Conventional disease-modifying antirheumatic drugs (cDMARDs) and bDMARDs targeting pro-inflammatory cytokines decrease inflammation and could thus improve CV risk, metabolic profile, and body composition [8]. However, randomized trials assessing the differential effects of DMARDs on body composition are lacking and the evolution of body composition under treatment is still controversial [9]. We investigated in a 1-year open follow-up study the effect of treatment with TNF inhibitor (TNFi) on body composition in biologic-naïve patients with active RA and compared to patients treated with conventional DMARDs (cDMARD) alone or with non-TNF-targeted biologics.

## 2. Experimental Section

### 2.1. Patients

Patients over 18 years-old with active RA who visited the Rheumatology Department of Clermont-Ferrand University Hospital for initiating first bDMARD were invited to participate to the longitudinal cohort RCVRIC analyzing cardiovascular risk and chronic inflammatory rheumatism (PHRC RCVRIC AOI 2014 N° ID-RCB-A01847-40). The patients fulfilled the 2010 RA classification criteria [10]. All patients were biologic-naive at baseline and the choice of treatment was left to the discretion of the rheumatologist. Only patients included between April 2014 and January 2019, with a follow-up of at least 1 year, at least one total body dual energy X-ray absorptiometry (DXA) at baseline and at 1 year, and who continued the same treatment for 1 year, either cDMARD or bDMARD, were analyzed. All patients had a longitudinal follow-up at 6 and 12 months. Three groups of patients were identified: patients treated with TNF inhibitor (TNFi); non-TNF-targeted biologic (non-TNFi) including tocilizumab, rituximab, abatacept; and controls (cDMARD alone). The study was approved by the local ethics committee of Clermont-Ferrand (Institutional Review Boards: AU 1161), and all the patients gave informed consent for participation.

We collected clinical and demographic data, disease and imaging characteristics, and cardio-metabolic profile of patients including weight, height, waist circumference, blood pressure, cholesterol-lowering, antihypertensive, and antidiabetic drugs. The duration, extra articular manifestations, the presence of rheumatoid factor and/or anti-CCP antibodies, and biological markers of inflammation (erythrocyte sedimentation rate (ESR; mm/h) and circulating concentration of C-reactive protein (CRP; mg/L)) were recorded. Disease activity was evaluated by the DAS 28ESR/CRP. Radiographic erosions were recorded on baseline feet and hands. Functional disability was evaluated with the health assessment (HAQ) and the Rheumatoid Arthritis Impact of Disease (RAID) score. Anxiety and depression were evaluated with the hospital anxiety and depression (HAD) questionnaire. We also recorded conventional DMARDs, steroids, and nonsteroidal anti-inflammatory drugs (NSAIDs).

### 2.2. Body Composition

All subjects underwent total body DXA scanning (HOLOGIC Discovery A S/N 85701). Fat, lean, and bone mass for the total body and per region (arms, legs, and trunk) were measured and analyzed using the manufacturer’s validated software (version 4.02 HOLOGIC APEX). Daily quality control and calibration procedures were performed using the manufacturer’s standard. Body fat percentage was calculated as the proportion of total fat mass to total mass. Appendicular fat and lean masses were computed as the sum of the tissue compartment (fat or lean) of both arms and legs. Skeletal muscle mass index (SMI) was calculated as appendicular lean mass divided by height^2^, fat mass index (FMI) as total fat mass divided by height^2^, and fat-free mass index (FFMI) as total body mass without total fat mass divided by height^2^. The trunk-peripheral fat ratio, a measure of “android” fat was calculated using fat of the body trunk divided by the peripheral (legs and arms) fat. Separation of subcutaneous adipose tissue (SAT) and visceral adipose tissue (VAT) were performed by two blinded readers inside a region of interest using a new software developed on DXA with a validated method [4].

### 2.3. Muscle Function, Physical Performance, and Sedentary Time

Muscle strength was assessed by the measure of the grip strength on the dominant hand (Handgrip). Physical performance was assessed using 6-min walk test. Sedentary time was assessed on the GPAQ questionnaire: “How much time do you usually spend sitting or reclining on a typical day?”.

### 2.4. Statistical Analysis

The sample size estimation was calculated according to effect-size bounds recommended by Cohen’s (Cohen, 1988): small (ES: 0.2), medium (ES: 0.5), and large (ES: 0.8, “‘grossly perceptible and therefore large”). More precisely, with a minimum of 45 TNFi with at 2–3 repeated measures for each patient, effect-size greater than 0.3 can be highlighted for a two-sided type I error at 0.05, a statistical power greater than 80% and an intraindividual correlation coefficient at 0.5. For example, for total lean mass, it corresponds to show a minimal difference between time points evaluation of 1 kg. For the secondary objectives aiming to compare patients treated with TNFi to cDMARDs alone or to non-TNF-targeted biologics, 48 patients in the group TNFi and 18 patients in each comparison groups allow to highlight an effect-size greater than 0.8, large enough to obtain preliminary data for secondary objectives, for a two-sided type I error at 0.05 and a statistical power greater than 80%.

Statistical analysis was performed using Stata software (version 15; StataCorp, College Station, TX, USA). All tests were two-sided, with a Type I error set at 0.05. Categorical parameters were expressed as frequencies and associated percentages, and continuous data as mean ± standard deviation or as median [interquartile range], according to statistical distribution. The three groups were compared at baseline with Chi-squared or Fisher’s exact tests for categorical variables and with ANOVA or Kruskal–Wallis test for quantitative ones. The Gaussian distribution was verified by the Shapiro–Wilk test and homoscedasticity by the Bartlett’s test. In order to evaluate whether the evolution of body composition and muscle function over time was different according to the three groups, mixed models were used considering the patient as random effect. Different models have been implemented depending on the nature of the dependent variable: linear mixed models in the case of quantitative variables and generalized linear mixed models with logit link function in the case of binary variables. The independent variables were the group (Control, TNFi, and Non-TNFi), the time (M0, M6, and M12) and their interaction, and the following baseline characteristics: age, rheumatoid arthritis duration, erosive rheumatoid arthritis, and glucocorticoids. Residuals normality of all models was studied, and a logarithmic transformation has been proposed when appropriate to achieve normality. Finally, body composition changes were measured by calculating variation rates between M0-M6 and M0-M12, and these rates were compared according to baseline characteristics with: Mann–Whitney test for binary characteristics (e.g., sex) and Spearman’s correlation coefficient for quantitative parameters (e.g., age). The relationship between significant change in body composition and variation for disease activity outcomes (EULAR response, HAQ, RAID, and CRP) was studied with correlation coefficients.

## 3. Results

### 3.1. Baseline Characteristics of the Study Participants

Of the 180 biologic-naïve RA patients included in the cohort between April 2014 and January 2019, with a follow-up of at least 1 year, 96 continued the same treatment. Of these patients, 83 had total body DXA at inclusion and 1 year, and it were analyzed (Figure 1).

The clinical and disease characteristics of the patients at baseline are presented in Table 1. The patients were mainly women (75%), with a mean age of 58.5 ± 10.8 years and median duration of RA of 3.7 years [1.5; 11.3]. Rheumatoid factor was present in 77% of the patients, anti-CCP in 78%, and erosions on radiographs in 44%. The disease was active (DAS28-ESR 4.2 ± 1.1). At inclusion, 86% of patients had methotrexate and 45% steroids with a median dose of 6.5 mg/day [5; 10]. TNFi was initiated in 47 patients (56.6%): 35 received etanercept, 6 adalimumab, and 1 golimumab. Eighteen patients (21.7%) received non-TNFi (10 patients treated with rituximab because others biological agents were contraindicated, 6 with abatacept, and 2 with tocilizumab). Finally, in 18 patients (21.7%), no bDMARD was initiated and they continued the cDMARD alone for 1 year. This named “control group” had shorter disease (*p* = 0.007) and less erosion on radiographs (*p* = 0.016). Steroids were more frequent in the non-TNFi group (*p* = 0.027).

Baseline body composition, muscle strength, and physical performance are presented in Table 2. Lean mass (total lean mass, FFMI, and SMI) and strength were higher in the control group even after adjustment on age, disease duration, erosion, and steroids, whereas no differences for fat mass were observed between the treatment groups.

### 3.2. Body Composition Changes According to the Treatment Received

No significant difference in body mass index (BMI) changes over time was observed whatever the treatment group. After 1 year of treatment with TNFi, lean mass as assessed by total lean mass (*p* = 0.015), FFMI (*p* = 0.013) and SMI (*p* = 0.010) increased significantly (Table 3). This increase mainly occurred between 6 and 12 months of treatment. The total body fat and the percentage of body fat were not changed but an increase in SAT (*p* = 0.042) was observed without change in VAT. These changes in body composition were associated with an improvement in handgrip strength (*p* < 0.001) as well as in walk test at 6 (*p* = 0.006) and 12 months (*p* < 0.001). After adjusting for the dose of corticosteroids at each visit, the significant changes observed for body composition and muscle function persisted.

For RA patients receiving non-TNFi (Table 3), no change in body composition was observed at 12 months. Significant changes were observed at 6 months, with an increase in body fat percentage (*p* = 0.016), a decrease in visceral fat (*p* = 0.024), and a decrease in total lean mass (*p* = 0.026). These changes were not associated with variation in muscle function.

RA patients treated with cDMARD alone (Table 3) did not show any significant change in body composition parameters except for an increase in the visceral fat at 12 months (*p* = 0.033). No change for muscle function or sedentary time was observed.

When the treatment groups were compared, no significant treatment × time effect was observed except a greater decrease in visceral fat observed in patients treated with non-TNFi only at 6 months compared to TNFi group. (Table 3).

### 3.3. Factors Associated with Body Composition Changes

Variations in body composition, muscle function, and sedentary time were analyzed according to response to treatment. After 1 year of treatment, good-moderate EULAR response was noted in 85% of patients with TNFi (*n* = 35/41), 83% with non-TNF biologic (*n* = 15/18) and 78% of patients (14/18) with cDMARD alone (controls). We did not observe significant differences for the change in body composition parameters according to the EULAR response. Change in handgrip strength at 12 months was different between EULAR (moderate or good) responders and nonresponders (*p* = 0.001) and 12 months (*p* = 0.001). The handgrip strength increased at 12 months (13.8 ± 9.8 to 20.1 ± 11.4 kg) in EULAR (moderate or good) responders, whereas it decreased in nonresponders (26.2 ± 14.0 to 22.6 ± 14.8 kg) (*p* = 0.001).

In RA patients treated with TNFi, lean mass changes were not correlated with baseline characteristics, including ESR, CRP, DAS28, HAQ, RAID, treatment with corticosteroids, anti-CCP positivity, and presence of erosion. The increase in lean mass at 12 months was also not correlated with changes for HAQ, RAID, or CRP. SAT increase after 12 months was more important in women (*p* = 0.04) and inversely correlated with the sedentary time at inclusion (*r* = −0.38, *p* = 0.030). Handgrip strength and walk tests improvements after 12 months were higher in patients with low baseline tests (handgrip *r* = −0.50, *p* = 0.001; and walk test *r* = −0.61 *p* < 0.001).

In the non-TNFi group, high baseline ESR and high RAID were associated with greater increase in total fat mass after 6 months (respectively, *r* = 0.54, *p* = 0.048 and *r* = 0.81, *p* = 0.045). Initial RAID was inversely correlated with the variation of total lean mass observed after 6 months (*r* = −0.86, *p* = 0.007).

No baseline factors were significantly associated with the increase in VAT at 12 months in patients with cDMARD alone.

## 4. Discussion

This study shows that treatment with TNFi for 1 year is associated with rather favorable changes of the body composition and muscle function in RA patients. The increase in lean mass was associated with an improvement in muscle strength and function. For the total lean mass after 12 months of TNFi treatment, a clinically relevant difference around 1 kg was highlighted, which corresponds to a mild effect-size equal to 0.38 (0.09; 0.66). If an increase in subcutaneous fat was noted, no change for BMI, total body fat, or visceral fat was observed. There were, however, no significant differences for body composition changes over time between treatment groups. Changes in body composition were not associated with EULAR response in contrast to muscle strength or walking ability but a low proportion of nonresponders could have been analyzed as the patients included had a stable treatment for at least 1 year. While the question of the specific effect of biologics on body composition has been raised for several years now, randomized trials are difficult to set up and still do not currently answer the question. Alterations in body composition (loss of muscle mass and increase in fat mass) occur early in RA [5] and are not completely reversible despite achieving remission or low activity [8,11]. Several studies have analyzed changes in body composition during treatment with TNFi in RA with conflicting results [12,13,14,15,16]. Only two studies were randomized [12,15]. Marcora et al. [12] compared 12 patients treated with etanercept and 14 patients with MTX. After 3 and 6 months of follow-up, they did not find any differences for body composition in the 2 treatment groups. Engvall et al. [15] evaluated the body composition in 18 patients treated with infliximab and MTX and compared to 22 patients treated by a combination of DMARDs (MTX, sulfasalazine, and hydroxychloroquine). They did not observe significant changes after 3 and 12 months of treatment but reported an increase in body fat mass after 2 years of follow-up (14 patients still in the TNFi group and 11 patients in the control group). Open labeled studies reported an increase in BMI and fat mass with variable muscle gain [13,14,16,17]. These data raise the question of the long-term metabolic tolerance profile of TNFi. However, the analysis of the distribution of the fat, subcutaneous and visceral adipose tissue, which have been identified as key actor in the development of cardiometabolic disorders, was lacking in previous studies. In addition, no studies assessed the simultaneous evolution of muscle function and performance (strength and walking) strongly associated with morbidity and mortality during ageing [18]. The increase in muscle mass and the stability of visceral adipose tissue observed in this study could suggest a favorable metabolic profile in patients treated with TNFi. The absence of insulin resistance assessment does not allow to conclude on the impact of muscle mass change on metabolism adaptation or the opposite. In addition to be a pro-inflammatory cytokine, IL6 is also a cytokine involved in energy metabolism and muscle function. It participates in glucose and lipid homeostasis by increasing lipolysis and oxidation of fatty acids. Thus, it stimulates visceral fat lipolysis when acutely produced, e.g., during physical exercise, and partly explains the benefits of physical activity on the risk of cardiometabolic diseases [19]. Conversely, chronic inflammation during RA is associated with persistent elevations of IL6 and insulin resistance and with an increased risk of metabolic syndrome even in patients of normal weight [20]. A few studies have investigated the effect of tocilizumab on body composition [4,21,22]. In a 1 year open follow-up study including 21 active RA patients [4], a significant increase in lean mass was observed without changes for body fat mass. Distribution of the fat was modified with a decrease in trunk/peripheral fat ratio and an increase in subcutaneous adipose tissue. Specific studies in the context of chronic inflammatory rheumatic diseases are, therefore, necessary to determine whether there is a differential effect between TNFi and IL6 inhibition on muscle, visceral adipose tissue, and glucose metabolism. No studies reporting the evolution of body composition with abatacept or rituximab was reported in RA. In patients with antineutrophil cytoplasmic antibody-associated vasculitis, a significant increase in BMI occurred during the first 6 months of treatment with rituximab and may be linked to improvement in disease activity and glucocorticoid exposure [23].

The main limitations of this study were the lack of randomization and the small sample size of the control and non-TNFi groups. However, in addition to previous open-labelled studies, our study allows parallel assessment of body composition (including subcutaneous and visceral fat mass) and muscle function, over a period of 1 year of stable treatment. Our study also highlights the impact of the disease on lean body mass and strength, especially in patients requiring biotherapy. Compared with patients treated with cDMARD alone, these patients were characterized by lower lean mass and strength, even after adjustment on disease duration, radiographic damage, and steroids, although they had similar disease activity. The increase in lean mass, muscle strength, and function during the follow-up could suggest a potential reversibility with TNFi. Less severe impairment in RA patients from the control group could also explain a smaller and nonsignificant gain at 1 year. In our study, the majority of the patients of the non-TNFi group were treated with rituximab (10/18). For this treatment, the efficacy period may be longer than for TNFi and may explain the increase in fat mass and decrease in lean mass noted at 6 months but not confirmed at 1 year. The small number of patients treated with abatacept and tocilizumab does not allow us to conclude on a specific effect of these treatments.

## 5. Conclusions

In this study, we observed altered body composition and muscle function in patients with active RA starting first biologic DMARD. In those patients, treatment with TNFi over 1 year was associated with an improvement in lean mass, strength, and physical performance without change for BMI, total body fat, or visceral fat suggesting a favorable metabolic profile of TNFi.

## Figures and Tables

**Figure 1 jcm-10-00487-f001:**
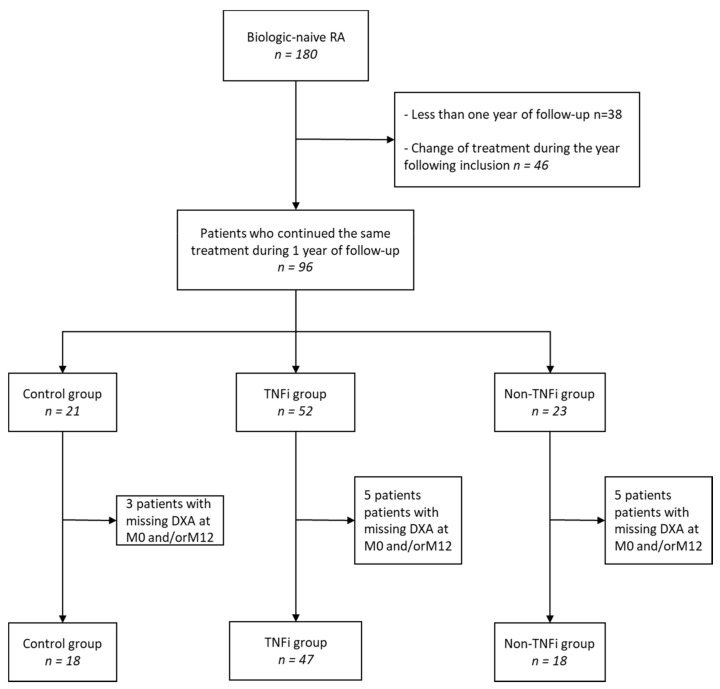
Flow of participants through study. RA: rheumatoid arthritis; DXA: dual energy X-ray absorptiometry

**Table 1 jcm-10-00487-t001:** Clinical and disease characteristics of the study participants at inclusion.

	Total (*n* = 83)	Control (*n* = 18)	TNFi (*n* = 47)	Non-TNFi (*n* = 18)	*p*
Age (years)	58.5 ± 10.8	57.9 ± 10.8	56.9 ± 11.2	63.4 ± 8.4	0.089
Female gender	62 (74.7)	11 (61.1)	37 (78.7)	14 (77.8)	0.42
Body mass index (kg/m^2^)	26.9 ± 6.0	29.0 ± 6.8	26.7 ± 6.4	25.3 ± 3.2	0.21
RA duration (years)	3.7 [1.5; 11.3]	0.3 [0.1; 3.8]	4.2 [1.6; 12.0]	5.4 [2.3; 14.8]	0.007
RF positive	64 (77.1)	13 (72.2)	34 (72.3)	17 (94.4)	0.15
Anti-CCP positive	65 (78.3)	15 (83.3)	34 (72.3)	16 (88.9)	0.32
Erosive RA	35/79 (44.3)	2/16 (12.5)	23/45 (51.1)	10 (55.6)	0.016
DAS28-ESR (*n* = 79)	4.21 ± 1.09	4.18 ± 1.22	4.24 ± 1.03	4.16 ± 1.17	0.96
DAS28-CRP (*n* = 80)	4.08 ± 1.06	3.89 ± 1.35	4.13 ± 0.90	4.14 ± 1.15	0.92
SDAI (*n* = 76)	20 [13; 27]	22 [15; 29]	19 [15; 27]	19 [13; 27]	0.80
CDAI (*n* = 76)	19 [13; 26]	21 [15; 29]	17 [13; 25]	19 [13; 26]	0.44
CRP (mg/L)	8 [3; 17]	6 [3; 17]	10 [3; 18]	6 [3; 23]	0.39
ESR (mm/h) (*n* = 81)	17 [9; 28]	17 [9; 25]	15 [9; 30]	18 [10; 34]	0.93
HAQ (*n* = 63)	0.8 [0.5; 1.3]	0.6 [0.4; 1.0]	0.8 [0.6; 1.3]	0.8 [0.4; 1.2]	0.50
VAS asthenia (mm) (*n* = 74)	54.3 ± 26.1	51.7 ± 24.0	56.1 ± 25.6	52.6 ± 30.2	0.78
RAID score (*n* = 61)	5.24 ± 1.93	4.70 ± 1.25	5.36 ± 2.01	5.54 ± 2.39	0.19
HAD-anxiety (*n* = 58)	9.4 ± 3.9	8.3 ± 3.4	9.8 ± 4.4	9.5 ± 2.6	0.23
HAD-depression (*n* = 59)	7.2 ± 3.6	5.7 ± 3.8	7.7 ± 3.5	7.4 ± 3.4	0.13
NSAIDs	18 (21.7)	4 (22.2)	11 (23.4)	3 (16.7)	0.94
Glucocorticoids	37 (44.6)	6 (33.3)	18 (38.3)	13 (72.2)	0.027
Methotrexate	71 (85.5)	16 (88.9)	41 (87.2)	14 (77.8)	0.69

Data are presented as frequencies (associated percentages), as mean ± standard deviation, or as median [interquartile range]. CCP: cyclic citrullinated peptide; CDAI: clinical disease activity index; CRP: C-reactive protein; DAS28: disease activity score in 28 joints; ESR: erythrocyte sedimentation rate; HAD: hospital anxiety and depression scale; HAQ: health assessment questionnaire; NSAIDs: nonsteroidal anti-inflammatory drugs; RA: rheumatoid arthritis; RAID: rheumatoid arthritis impact of disease; RF: rheumatoid factor; SDAI: simple disease activity index; TNFi: tumor necrosis factor inhibitor; VAS: visual analogue scale.

**Table 2 jcm-10-00487-t002:** Baseline body composition, muscle strength, and physical performance of rheumatoid arthritis patients.

	Total (*n* = 83)	Control (*n* = 18)	TNFi (*n* = 47)	Non-TNFi (*n* = 18)	*p*	*p* ^a^
Total fat mass (kg)	24.4 ± 11.5	26.5 ± 12.8	25.3 ± 12.2	20.2 ± 7.1	0.19	0.33
Total fat mass (%)	31.5 ± 8.9	30.2 ± 10.9	32.8 ± 8.3	29.6 ± 7.9	0.34	0.33
FMI (kg/m^2^)	9.3 ± 4.7	9.5 ± 5.3	9.4 ± 4.5	8.8 ± 4.9	0.85	0.78
Trunk/peripheral FM ratio	0.92 ± 0.28	1.00 ± 0.34	0.88 ± 0.22	0.95 ± 0.34	0.42	0.61
VAT (cm^2^)	100 [51; 159]	141 [80; 179]	90 [51; 137]	98 [51; 139]	0.28	0.28
SAT (cm^2^)	288 [212; 408]	317 [243; 494]	291 [212; 449]	670 [196; 348]	0.49	0.57
VAT/SAT ratio	0.30 [0.24; 0.46]	0.32 [0.25; 0.67]	0.29 [0.23; 0.36]	0.30 [0.24; 0.51]	0.30	0.15
Total lean mass (kg)	51.2 ± 12.1	58.7 ± 13.7	49.6 ± 10.8	47.7 ± 11.0	0.009	0.012
FFMI (kg/m^2^) (*n* = 82)	18.5 ± 3.0	20.2 ± 3.3	18.1 ± 2.9	17.8 ± 2.1	0.037	0.050
SMI (kg/m^2^)	7.9 ± 1.5	8.7 ± 1.7	7.7 ± 1.4	7.5 ± 1.1	0.026	0.042
FMI/SMI ratio	1.18 ± 0.52	1.11 ± 0.61	1.21 ± 0.48	1.18 ± 0.55	0.63	0.82
Handgrip strength (kg) (*n* = 70)	15.9 ± 11.4	23.7 ± 13.6	13.5 ± 8.9	13.8 ± 12.1	0.023	0.017
6 MWT (meters) (*n* = 64)	455 [373; 540]	485 [420; 511]	435 [370; 540]	490 [370; 540]	0.83	0.78
Sedentary (h/week) (*n* = 57)	42 [21; 70]	28 [21; 56]	42 [21; 84]	42 [28; 53]	0.50	0.69

Data are presented as mean ± standard deviation or as median [interquartile range]. FFMI: fat-free mass index; FM: fat mass; FMI: fat mass index; SAT: subcutaneous adipose tissue; 6 MWT: 6-min walk test; SMI: skeletal muscle mass index; TNFi: tumor necrosis factor inhibitor; VAT: visceral adipose tissue. *p*: unadjusted comparisons; *p*
^a^: adjusted comparisons on baseline characteristics: age, rheumatoid arthritis duration, erosive rheumatoid arthritis, and glucocorticoids.

**Table 3 jcm-10-00487-t003:** Evolution of body composition and muscle function after 6 and 12 months of treatment.

	Control	*p* ^C^	TNFi	*p* ^T^	Non-TNFi	*p* ^N^	*p* ^CT^	*p* ^CN^	*p* ^TN^
Body mass index (kg/m^2^)									
M0 (*n* = 18/47/18)	29.0 ± 6.8		26.7 ± 6.4		25.3 ± 3.2				
M6 (*n* = 18/46/18)	29.0 ± 7.1	0.67	26.9 ± 6.5	0.71	25.3 ± 3.2	0.69	0.92	0.99	0.91
M12 (*n* = 18/47/18)	29.0 ± 6.9	0.88	27.0 ± 6.5	0.12	25.2 ± 3.3	0.97	0.35	0.95	0.38
Total fat mass (kg)									
M0 (*n* = 18/47/18)	26.5 ± 12.8		25.3 ± 12.2		20.2 ± 7.1				
M6 (*n* = 16/44/14)	26.0 ± 12.8	0.27	26.6 ± 12.5	0.26	21.6 ± 7.4	0.069	0.65	0.49	0.17
M12 (*n* = 18/47/18)	27.0 ± 12.6	0.22	25.4 ± 11.5	0.77	20.6 ± 7.4	0.73	0.33	0.58	0.84
Total fat mass (%)									
M0 (*n* = 18/47/18)	30.2 ± 10.9		32.8 ± 8.3		29.6 ± 7.9				
M6 (*n* = 16/44/14)	29.8 ± 9.3	0.47	34.0 ± 8.0	0.36	32.6 ± 6.5	0.016	0.96	0.14	0.075
M12 (*n* = 18/47/18)	30.7 ± 10.1	0.55	32.5 ± 8.4	0.60	30.1 ± 7.9	0.57	0.48	0.94	0.43
FMI (kg/m^2^)									
M0 (*n* = 18/47/18)	9.5 ± 5.3		9.4 ± 4.5		8.8 ± 4.9				
M6 (*n* = 16/44/14)	9.2 ± 4.8	0.46	9.9 ± 4.7	0.11	8.6 ± 2.6	0.59	0.65	0.57	0.58
M12 (*n* = 18/47/18)	9.7 ± 5.3	0.42	9.5 ± 4.4	0.57	7.9 ± 2.8	0.31	0.31	0.18	0.20
Trunk/peripheral FM ratio									
M0 (*n* = 18/47/18)	1.00 ± 0.34		0.88 ± 0.22		0.95 ± 0.34				
M6 (*n* = 16/44/14)	1.02 ± 0.35	0.17	0.91 ± 0.23	0.16	0.95 ± 0.30	0.19	0.33	0.75	0.55
M12 (*n* = 18/47/18)	1.05 ± 0.31	0.12	0.90 ± 0.24	0.12	0.96 ± 0.30	0.62	0.27	0.33	0.76
VAT (cm^2^)									
M0 (*n* = 18/47/18)	141 [80; 179]		90 [51; 137]		98 [51; 139]				
M6 (*n* = 16/44/14)	149 [45; 177]	0.40	94 [58; 146]	0.93	81 [51; 140]	0.024	0.55	0.18	0.043
M12 (*n* = 18/47/18)	157 [72; 187]	0.033	92 [61; 133]	0.99	74 [58; 147]	0.99	0.16	0.21	0.99
SAT (cm^2^)									
M0 (*n* = 18/47/18)	317 [243; 494]		291 [212; 449]		270 [196; 348]				
M6 (*n* = 16/44/14)	296 [244; 451]	0.16	324 [244; 467]	0.003	250 [212; 394]	0.31	0.61	0.97	0.61
M12 (*n* = 18/47/18)	323 [219; 522]	0.15	323 [224; 466]	0.042	265 [199; 369]	0.74	0.97	0.26	0.17
VAT/SAT ratio									
M0 (*n* = 18/47/18)	0.32 [0.25; 0.67]		0.29 [0.23; 0.36]		0.30 [0.24; 0.51]				
M6 (*n* = 16/44/14)	0.32 [0.21; 0.59]	0.80	0.29 [0.20; 0.37]	0.45	0.28 [0.24; 0.36]	0.052	0.98	0.16	0.017
M12 (*n* = 18/47/18)	0.34 [0.28; 0.55]	0.95	0.28 [0.23; 0.37]	0.91	0.34 [0.21; 0.52]	0.50	0.98	0.63	0.44
Total lean mass (kg)									
M0 (*n* = 18/47/18)	58.7 ± 13.7		49.6 ± 10.8		47.7 ± 11.0				
M6 (*n* = 16/44/14)	58.0 ± 13.2	0.53	49.5 ± 10.7	0.63	43.3 ± 7.4	0.026	0.50	0.21	0.053
M12 (*n* = 18/47/18)	59.0 ± 14.5	0.76	50.7 ± 11.3	0.015	47.4 ± 10.9	0.49	0.26	0.48	0.062
FFMI (kg/m^2^)									
M0 (*n* = 18/47/17)	20.2 ± 3.3		18.1 ± 2.9		17.8 ± 2.1				
M6 (*n* = 16/44/14)	20.0 ± 3.2	0.57	18.2 ± 2.9	0.61	17.3 ± 2.0	0.072	0.52	0.38	0.16
M12 (*n* = 18/47/18)	20.4 ± 3.5	0.50	18.5 ± 3.2	0.013	18.0 ± 2.2	0.75	0.35	0.79	0.23
SMI (kg/m^2^)									
M0 (*n* = 18/47/18)	8.7 ± 1.7		7.7 ± 1.4		7.5 ± 1.1				
M6 (*n* = 16/44/14)	8.7 ± 1.4	0.87	7.7 ± 1.4	0.47	7.2 ± 1.0	0.12	0.62	0.39	0.092
M12 (*n* = 18/47/18)	8.8 ± 1.8	0.42	7.9 ± 1.5	0.010	7.5 ± 1.1	0.83	0.65	0.44	0.12
FMI/SMI ratio									
M0 (*n* = 18/47/18)	1.11 ± 0.61		1.21 ± 0.48		1.18 ± 0.55				
M6 (*n* = 16/44/14)	1.06 ± 0.56	0.94	1.27 ± 0.47	0.39	1.21 ± 0.38	0.96	0.69	0.92	0.66
M12 (*n* = 18/47/18)	1.12 ± 0.58	0.89	1.20 ± 0.47	0.68	1.08 ± 0.40	0.37	0.90	0.44	0.28
Handgrip strength (kg)									
M0 (*n* = 16/41/13)	23.7 ± 13.6		13.5 ± 8.9		13.8 ± 12.1				
M6 (*n* = 15/39/16)	23.6 ± 11.3	0.36	17.8 ± 9.5	<0.001	17.4 ± 12.8	0.42	0.34	0.82	0.21
M12 (*n* = 16/43/14)	26.6 ± 12.5	0.22	20.2 ± 10.6	<0.001	15.9 ± 12.7	0.15	0.19	0.90	0.15
6 MWT (meters)									
M0 (*n* = 14/35/15)	485 [420; 511]		435 [370; 540]		490 [370; 540]				
M6 (*n* = 13/37/14)	482 [425; 560]	0.86	477 [417; 550]	0.006	535 [445; 606]	0.067	0.15	0.18	0.88
M12 (*n* = 16/44/14)	495 [449; 597]	0.11	508 [459; 562]	<0.001	483 [428; 564]	0.36	0.87	0.61	0.081
Sedentary (h/week)									
M0 (*n* = 13/33/11)	28 [21; 56]		42 [21; 84]		42 [28; 53]				
M6 (*n* = 13/27/11)	28 [14; 42]	0.45	42 [21; 63]	0.43	28 [28; 42]	0.84	0.66	0.65	0.79
M12 (*n* = 12/25/9)	35 [21; 63]	0.41	42 [21; 70]	0.52	42 [28; 53]	0.36	0.31	0.99	0.32

Data are presented as mean ± standard deviation or as median [interquartile range]. All comparisons are adjusted on baseline characteristics: age, rheumatoid arthritis duration, erosive rheumatoid arthritis, and glucocorticoids. FFMI: fat-free mass index; FM: fat mass; FMI: fat mass index; SAT: subcutaneous adipose tissue; 6 MWT: 6-min walk test; SMI: skeletal muscle mass index; TNFi: tumor necrosis factor inhibitor; VAT: visceral adipose tissue. *p*^C^: subgroup analysis (Control) compared to M0; *p*^T^: subgroup analysis (TNFi) compared to M0; *p*^N^: subgroup analysis (Non-TNFi) compared to M0; *p*^CT^: interaction between time (compared to M0) and group (TNFi compared to Control); *p*^CN^: interaction between time (compared to M0) and group (Non-TNFi compared to Control); *p*^TN^: interaction between time (compared to M0) and group (Non-TNFi compared to TNFi).

## Data Availability

The data presented in this study are available on request from the corresponding author.

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
