# Peer review of "The Effect of TNF and Non-TNF-Targeted Biologics on Body Composition in Rheumatoid Arthritis"

_jcm, 2021, doi:10.3390/jcm10030487_

Round 1
Reviewer 1 Report
This manuscript is about the effect of anti-TNF on body composition in Ra patients. Authors presented that anti-TNF treatment is associated with increase of total lean mass and fat free mass index, and muscle function. However, there are some points clarified.
- Chronic inflammatory disease such as RA is associated with significant metabolic derangement including defects of glycolysis, shunting to PPP. Thus improvement of inflammation may be linked to the muscle improvement. Authors need to discuss the reason why anti-TNF only improve muscle mass and muscle function.
- Authors need to show what anti-TNFs were used for patients.
- Additionally, patients treated with rituximab may have prior history of non-responsiveness or intolerance to anti-TNF agent. Please identify the prior results with anti-TNFs. The poor response to anti-TNF may affect the response to non-TNF biologics in body composition. How did you authors treat these effects associated with prior use of anti-TNFs?
- Clinical response was described only with EULAR response. Could you tell us about the relationship between other disease activity outcomes and body composition.
- What about the change of corticosteroid dose after biologic agent? I wonder if the decrease of corticosteroid dose may affect clinical outcome related with body composition.
Author Response
Dear Editor,
Thank you for allowing us to resubmit our manuscript with revision required for publication.
We would like to thank the reviewers for their contribution and comments that allowed us to improve the quality of our manuscript. You will find below the detailed answers to the comments.
Yours sincerely,
Pr Anne Tournadre
Reviewer 1. This manuscript is about the effect of anti-TNF on body composition in Ra patients. Authors presented that anti-TNF treatment is associated with increase of total lean mass and fat free mass index, and muscle function. However, there are some points clarified.
- Chronic inflammatory disease such as RA is associated with significant metabolic derangement including defects of glycolysis, shunting to PPP. Thus improvement of inflammation may be linked to the muscle improvement. Authors need to discuss the reason why anti-TNF only improve muscle mass and muscle function.
Response : Thank you for this comment. The improvement in muscle mass and stability of visceral mass could suggest a favourable metabolic profile in patients treated with TNFi but insulin resistance was not assessed in this study. It is then difficult to conclude on the impact of muscle mass change on metabolism adaptation or the opposite. It is also difficult to conclude on the impact of specific cytokine signalling (TNF or IL6) on insulin resistance commonly associated with RA but we added a discussion on the specific effect of IL6 on the glucose and lipid metabolisms.
- Authors need to show what anti-TNFs were used for patients.
Response : Among the 47 patients treated with TNFi, 35 received etanercept, 6 adalimumab and 1 golimumab. This was specified in the manuscript.
- Additionally, patients treated with rituximab may have prior history of non-responsiveness or intolerance to anti-TNF agent. Please identify the prior results with anti-TNFs. The poor response to anti-TNF may affect the response to non-TNF biologics in body composition. How did you authors treat these effects associated with prior use of anti-TNFs?
Response : All patients included in this study were biologic-naïve, this is specified in the method section and flow-chart. Rituximab was used as first biologic in 10 patients because other biological agents were contraindicated.
- Clinical response was described only with EULAR response. Could you tell us about the relationship between other disease activity outcomes and body composition.
Response : Thank you for this suggestion. Correlations between significant change in body composition and disease characteristics including ESR, CRP, DAS28-ESR, DAS28-CRP, HAQ, RAID, treatment with corticosteroids, anti-CCP positivity and presence of erosion were analysed at baseline. For disease outcomes, the relationship between body composition and variation for EULAR response which takes into account both baseline and DAS28 changes was studied. As suggested by the reviewer, we added correlation analysis for others clinical/disease outcomes such as HAQ, RAID, CRP. This was added in the method and results sections.
- What about the change of corticosteroid dose after biologic agent? I wonder if the decrease of corticosteroid dose may affect clinical outcome related with body composition.
Response: Thanks a lot to give us the opportunity to clarify this important point concerning corticosteroids. At inclusion in the study, 37 patients were treated with corticosteroids with a median dose of 6.5 mg/day [5; 10]. At 6 and 12 months, 70 % (26/37) and 59 % (22/37), still received corticosteroids with a median dose 5 mg/day [3; 5]. After adjusting for the dose of corticosteroids, the significant changes observed for body composition in patients treated with TNFi persisted: SAT at M6 p=0.001; SAT at M12 p=0.01; total lean mass at M12 p=0.05; FFMI at M12 p=0.04; SMI at M12 p=0.03; handgrip M6 p=0.04; handgrip M12 p<0.001; 6 MWT M6 p=0.01; 6 MWT p<0.001.
Reviewer 2 Report
Overall it was an interesting and up- to date work to read on an important topic. This study shows that treatment with TNFi for one year is associated with rather favourable changes of the body composition and muscle function in RA patients. The increase in lean mass was associated with an improvement in muscle strength and function
minor comments:
1. Can you please describe the matching process in detail? Was the sample size calculation performed for each group?
2. In Factors associated with body composition changes. Have they analyzed if damage measures in RA i. e the presence of erosions, the mean Health Assessment Questionnaire (HAQ) and anticitrullinated protein antibody, were predictors of changes of the body composition?
Author Response
Dear Editor,
Thank you for allowing us to resubmit our manuscript with revision required for publication.
We would like to thank the reviewers for their contribution and comments that allowed us to improve the quality of our manuscript. You will find below the detailed answers to the comments.
Yours sincerely,
Pr Anne Tournadre
Reviewer 2. Overall it was an interesting and up- to date work to read on an important topic. This study shows that treatment with TNFi for one year is associated with rather favourable changes of the body composition and muscle function in RA patients. The increase in lean mass was associated with an improvement in muscle strength and function
minor comments:
- Can you please describe the matching process in detail? Was the sample size calculation performed for each group?
Response : No matching were performed. This study is an open follow-up study as specified at the end of the introduction “We investigated in a 1-year open follow-up study the effect of treatment with TNF inhibitor (TNFi) on body composition in biologic-naïve patients with active RA and compared to patients treated with conventional DMARDs (cDMARD) alone or with non-TNF-targeted biologics.” In the method section, patients “All patients were biologic-naive at baseline and the choice of treatment was left to the discretion of the rheumatologist.”
As described in the manuscript, sample size was estimated to guaranty a satisfactory statistical power for the primary objective of this study, i.e. evolution of body composition with TNFi treatment. A minimum of 45 TNFi with at 2 to 3 repeated measures for each patient was necessary to describe an effect-size greater than 0.3 for a two-sided type I error at 0.05, a statistical power greater than 80% and an intra-individual correlation coefficient at 0.5. For the secondary objectives aiming to compare patients treated with TNFi to cDMARDs alone or to non-TNF-targeted biologics, with 48 patients in the group TNFi, it was necessary to include 18 patients in each comparison groups in order to highlight an effect-size greater than 0.8, large enough to obtain preliminary data for secondary objectives, for a two-sided type I error at 0.05 and a statistical power greater than 80%.
- In Factors associated with body composition changes. Have they analyzed if damage measures in RA i. e the presence of erosions, the mean Health Assessment Questionnaire (HAQ) and anticitrullinated protein antibody, were predictors of changes of the body composition?
Response : Correlations between significant change in body composition and baseline disease characteristics including ESR, CRP, DAS28-ESR, DAS28-CRP, HAQ, RAID, treatment with corticosteroids, anti-CCP positivity and presence of erosion were analyzed. This was more precisely specified in paragraph 3.3. Only significant correlations were specified in the results.